# Two Ways of Targeting a CD19 Positive Relapse of Acute Lymphoblastic Leukaemia after Anti-CD19 CAR-T Cells

**DOI:** 10.3390/biomedicines11020345

**Published:** 2023-01-25

**Authors:** Audrey Grain, Jocelyn Ollier, Thierry Guillaume, Patrice Chevallier, Baptiste Le Calvez, Marion Eveillard, Béatrice Clémenceau

**Affiliations:** 1Nantes Université, Inserm UMR 1307, CNRS UMR 6075, Université d’Angers, CRCI2NA, F-44007 Nantes, France; 2Paediatric Haematology and Oncology Nantes University Hospital, 44093 Nantes, France; 3Haematology Department, Nantes University Hospital, 44093 Nantes, France; 4Haematology Biology Department, Nantes University Hospital, 44093 Nantes, France

**Keywords:** acute lymphoblastic leukaemia, immunotherapy, relapse, CAR-T cells

## Abstract

Background: Therapeutic options for CD19^+^ relapses after anti-CD19 CAR-T cells are still debated; second infusion of anti-CD19 CAR-T cells, therapeutic antibodies, or targeted therapies can be discussed. Here, we explore the immunophenotyping and lysis sensitivity of CD19^+^ ALL relapse after anti-CD19 CAR-T cells and propose different therapeutic options for such a high-risk disease. Methods: Cells from successive B-ALL relapses from one patient were collected. A broad immunophenotype analysis was performed. ^51^Cr cytotoxic assays, and long-term killing assays were conducted using T-cell effectors that are capable of cytotoxicity through three recognition pathways: antibody-dependent cell-mediated cytotoxicity (ADCC), anti-CD19 CAR-T, and TCR. Results: Previously targeted antigen expression, even if maintained, decreased in relapses, and new targetable antigens appeared. Cytotoxic assays showed that ALL relapses remained sensitive to lysis mediated either by ADCC, CAR-T, or TCR, even if the lysis kinetics were different depending on the effector used. We also identified an immunosuppressive monocytic population in the last relapse sample that may have led to low persistence of CAR-T. Conclusion: CD19^+^ relapses of ALL remain sensitive to cell lysis mediated by T-cell effectors. In case of ALL relapses after immunotherapy, a large immunophenotype will make new therapies possible for controlling such high risk ALL.

## 1. Introduction

T-cells expressing a chimeric antigen receptor (CAR) recognizing the CD19 antigen have shown high early response rates in high-risk B acute lymphoblastic leukaemia (B-ALL) [1]. Nevertheless, 35% to 44% of patients relapse, and more than 50% of them present CD19^+^ relapses [2,3]. CD19^+^ relapses are often related to low potency or low persistence of CAR-T cells [4]. Amplification and persistence of CAR-T cells may be negatively impacted by the choice of co-stimulatory domain, T-cell subtype composition, and an unfavourable immune environment [4,5]. Therapeutic options are still debated in the CD19^+^ relapse setting: a second infusion of anti-CD19 CAR-T or antibodies targeting ALL have been proposed [6]. Humanised anti-CD19 CAR-T or associating a second infusion of anti-CD19 CAR-T cells with an anti-PD1 are other strategies explored in on-going Phase 1 studies [4,7,8,9]. 

To our knowledge, the susceptibility of ALL relapses to T-cells mediated lysis has not been explored so far. Here, we analyse broad immunophenotyping and cytotoxicity assays, conducted on ALL cells from successive relapses in one patient after three different immunotherapies in order to consider the optimal therapeutic way in this particularly complex setting. 

## 2. Materials and Methods

### 2.1. Case Description

A 21-year-old patient presented with a hyperleukocytic CRLF2 overexpressing and IKZF1 deleted B-ALL (Dg) in November 2019. After failed induction chemotherapy combined with rituximab, intensification led to the first complete remission (CR) with positive minimal residual disease of 0.3% by multicolour flow cytometry (MFC) and 7 × 10^−3^ by molecular biology (based on Ig/TCR gene rearrangement). A geno-identical allogeneic HSCT was then performed. A relapse (R2) occurred 2 months later, and the patient received salvage chemotherapy followed by anti-CD19 CAR-T cells (tisagenlecleucel, 1.9 × 10^9^ total viable cells, of which 9.4% were CAR^+^ (based on the manufacturer’s certificate) in July 2020. Bone marrow aspiration showed 81% of blasts before anti-CD19 CAR-T infusion. Circulating CAR-T cells in blood were detected using multicolour flow cytometry (MFC) and showed an expansion peak at day 8 (3.5% of T lymphocytes). Interestingly, the contemporary appearance of CD14^+^/HLA-DR^lo/neg^ monocytic cells was noted. A second CR was obtained with negative minimal residual disease at 1 month post-anti-CD19 CAR-T, persisting at 3 months post-infusion. The patient presented a loss of B cell aplasia at 4.5 months and a CD19^+^ relapse at 5.5 months post-CAR-T cells (R3). He received palliative chemotherapy and died in January 2022 (Figure 1). The patient provided signed consent for cell collection. 

### 2.2. Cells

At diagnosis, peripheral blood cells were collected from the leukapheresis product. Peripheral blood samples were collected at relapses R2 and R3. All contained at least 90% of blastic cells. Peripheral blood mononucleated cells (PBMC) were then isolated using density gradient centrifugation on a FICOLL-Paque solution (Eurobio). Cells were then frozen and used for experiments after thawing. The results could thus have been compared between relapses and initial diagnosis. 

#### 2.2.1. Cell Lines

Epstein-Barr B-lymphoblastoid cell lines (BLCL) were used as controls in all experiments and were derived from donor peripheral-blood mononuclear cells (PBMCs) following in vitro infection with EBV containing culture supernatant from the Marmoset B95-8 cell line in the presence of 1 µg/mL cyclosporin-A (sandimmun).

#### 2.2.2. Effector Cells

The patient’s anti-CD19 CAR-T cells were collected from residual material in the bag after infusion of tisagenlecleucel. We performed an independent analysis of the cells in our laboratory. Of the CD3^+^ cells, 11% were CAR^+^. This result was close to that reported in the manufacturer certificate. The CD4^+^/CD8^+^ ratio among CAR^+^ cells was 3.4 (77.5% CD4^+^ and 22.3% CD8^+^). CAR-T cells were largely composed of a differentiated T-cell subset, with CD62L^+^, CCR7^−^, and CD45RA^−^ phenotypes.Because the tisagenlecleucel preparation only contained 9% of anti-CD19-CAR^+^ T-cells, the anti-CD19-CAR^+^ cells were selected with a CD19 biotin-coupled protein (Miltenyi Biotec). An APC-coupled anti-biotin antibody was then used for FACS-sorting the anti-CD19-CAR^+^ cells. This T-cell population containing 95% of anti-CD19-CAR^+^ T-cells (of which 71% were CD8^+^ and 12% were CD4^+^) was used for the cytotoxic assays. Note that the remaining 7% of CD4^-^ and CD8^−^ cells were composed of 72% of αβ-T lymphocytes and 28% of Υδ-T lymphocytes.The anti-HLA-DPB1*04:01 CD4^+^ T-clone has been previously described [10]. Cells were grown in RPMI 1640 culture medium (Eurobio) supplemented with 8% human serum, 300 IU/mL IL-2, 2 mM l-glutamine, penicillin, and streptomycin (Gibco).An anti-CMVpp65 polyclonal CD8^+^-T-cell population was obtained as previously described [11] and grown in the same culture media.A CD8^+^ polyclonal T cell population transduced by a retroviral vector expressing a chimeric-receptor containing the murine CD16 receptor murine FcɣRIII, linked to the human-chain FcɛRIɣ, was obtained as previously described and used to perform antibody-dependent cell-mediated cytotoxicity (ADCC) assays with murine antibodies [12].

See all the effector cells in Appendix A.

### 2.3. Immunophenotype

Extended immunophenotyping of Dg, R2, and R3 cells was performed using the Human Cell Surface Marker Screening Kit from Biolegend^®^ ((Biolegend Europe B.V, Amsterdam, The Netherlands) (LEGENDscreen). The Human Cell Surface Marker Screening Kit contains 4 96-wells plates. Each well contains a PE-coupled antibody targeting one human surface antigen. Cells were first incubated with an anti-CD19-APC coupled antibody (Biolegend^®^, clone HIB19) in order to select the leukaemic blasts by gating CD19^+^ cells for analysis. A minimum of 10 000 CD19^+^ cells was acquired on a CANTO II cytometer (BD Biosciences, Le Pont de Claix, France) (Plateforme CytoCell–Nantes University), and the results were analysed using FlowJo v10.8.1 software (BD LifeSciences). Results were given in Relative Fluorescence Intensity (RFI in log) which is calculated as median fluorescence intensity of the targeted-antigen coupled antibody/median fluorescence intensity of the unspecific control isotype. All antigens screened in the panel are detailed in Appendix A.

### 2.4. Cytotoxicity Assays

#### 2.4.1. ^51.^ Cr Assays

In order to assess sensitivity to T-cells-induced lysis of each ALL sample, using different way of recognition (TCR, ADCC, CAR), standard 51Cr assay was first performed. Target cells were previously labeled with 75 µCi 51Cr for 1 h at 37 °C then washed 4 times with RPMI, SVF 10%. For cytotoxic assay using anti-CMV-pp65 CD8^+^ T cells, target cells were previously incubated for 30 min at 37 °C with 1µg/mL final concentration of CMVpp65 peptides pool (PepTivator CMV pp65 human) (Miltenyi Biotec SAS, Paris, France) and then washed twice. Target cells and effector T cells were then plated at the indicated effector-to-target ratio (E:T ratio) in flat-bottom 96 wells plates. A BLCL line was used as control. After a 4 h incubation at 37 °C, 25 μL of supernatant was removed from each well, mixed with 100 μL scintillation fluid (Ultima Gold XR) (PerkinElmer Health Sciences, Groninger, The Netherlands), and released 51Cr activity was counted in a scintillation counter (MicroBeta JET) (PerkinElmer Health Sciences, Groninger, The Netherlands). Each test was performed in triplicate. Results are expressed as the percentage of lysis, which is calculated according to the following equation: (experimental release − spontaneous release)/ (maximal release − spontaneous release) × 100. Three experiments in triplicate were performed independently. 

##### 2.4.2. Long-Term Killing Assays

Sensitivity to CAR-T induced lysis was analysed over 24 h because it better represents in vivo CAR-T activity [13]. Target (200,000 cells/well) and effector cells were plated at the E:T ratio 3:1 in a flat-bottom 96 wells plate and incubated at 37 °C. At different time points: 0 h (H0), 4 h (H4) and 24 h (H24) after co-culture, cell suspension was collected and wells were rinsed in order to collect all residual cells. Cells were then washed with PBS EDTA (0.02%) and with PBS before being labelled with a Fixable Viability Stain 780 (BD Biosciences) over 10 min at room temperature (RT). After having been washed twice, cells were then incubated with both a PE-coupled anti-CD22 antibody (Dako product provided by Technologies Agilent France, Les Ulis, France) and a FITC coupled anti-CD3 antibody (Beckman Coulter France SAS, Roissy, France) over 15 min RT, to distinguish effector and target residual viable cells. After two more washing with PBS-0.1% human albumin, cells were fixed before analysis by MFC. For each condition, the % of viable leukaemic cells (VS780^low^/CD22^+^) at time 0 was reported as 100%. Thus, at time 4 h and 24 h, the percentage of residual viable leukaemic cells was calculated as follows: = (% of viable CD22+ cells at this time/ % of viable CD22+ cells at H0) × 100.

Two experiments in duplicate were performed independently. All MFC analysis was performed on BD FACSCanto II (Plateforme CytoCell – Nantes University). 

## 3. Results

### 3.1. ALL Immunophenotypes

Extended immunophenotyping showed that the level of expression of CD19 decreased from a Relative Fluorescence Intensity (RFI) of 171 at Dg, to 13 and 23 at R2 and R3, respectively. Expression of the previously targeted CD20 also decreased from Dg (RFI = 36) to R2 (RFI = 12) and R3 (RFI = 13). In contrast, CD22 was highly expressed in all samples.

Interestingly, while CD135 (FLT3) expression was present on only 7% of the Dg ALL cells, at relapse R2 its expression was observed on all the cells and was maintained at relapse R3. Similarly, CD268 (BAFF-R), CD304 (Neuropilin-1), and CD71 expressions increased on the successive relapse cells.

Expression of molecules involved in immunological synapse and co-stimulatory ligands remained stable, and expression of checkpoint inhibitors was low in all samples (Appendix A).

The phenotypic analysis also revealed the presence of a monocytic CD14^+^, HLA-DR^neg^, CD33^+^, and CD11b^high^ population in the R3 relapse sample (3.8% of cells), and we showed that this population was detectable in the patient’s peripheral blood 7 days after CAR-T cell infusion. The immunophenotype of these monocytic cells is presented in Appendix A. 

### 3.2. Lysis Sensitivity Mediated by T-Cells Using Different Recognition Pathways: Anti-CD19 -CAR, ADCC, and TCR:

#### 3.2.1. The Anti-CD19 CAR Pathway 

Four-hour ^51^Cr cytotoxicity assays were performed, analysing the three ALL sample lyses induced by the FACS of CD19-CAR^+^ cells. All ALL samples were efficiently and similarly lysed by the patient’s anti-CD19 CAR-T cells, even R3 (Figure 2a). The lysis scores were directly proportional to the effector–target ratio. The rapid lysis of all the ALL through the CAR recognition pathway was confirmed by the long-term killing assays. Only 0.5% of the residual viable CD22^+^ cells were seen after 4 h of co-culture (Figure 3b). The percentage of residual viable cells in the wells where ALL cells were cultured alone are presented as control (Figure 3a).

#### 3.2.2. The ADCC Pathway 

Four antigens, highly expressed by the three ALL (Dg, R2 and R3) were selected for the ADCC assays: CD19 (clone HIB19), CD24 (adhesion molecule, in B cells; clone ML5), CD184 (IL-4 receptor; clone 12G5), and CD220 (insulin receptor; clone B6.220). The mCD16-T lymphocytes were used as effector cells. For all targeted antigens, ADCC lysis scores ranged from 20% to 35% within four-hour ^51^Cr cytotoxicity assays. In addition, ALL sensitivity to ADCC-mediated cell lysis remained stable from Dg to successive relapses, whatever antigen was targeted. No cell lysis was observed in the absence of antibodies (Figure 2d). Because of the lack of biological material, long-term killing assays could not be performed for ADCC. 

#### 3.2.3. The TCR Pathway

In the ^51^Cr cytotoxicity assay, little or no ALL cell lysis was observed after co-culture with the anti-HLA DPB1*04:01 CD4^+^ T-clone or the anti-CMV^pp65^ CD8-polyclonal T-cells (Figure 2b,c). However, in the long-term killing assays, after 24 h of co-culture, the percentage of residual leukaemic viable CD22^+^ cells progressively fell to 30% in the presence of anti-CMV^pp65^ CD8-T cells and to 35% in the presence of the anti-HLA DPB1*04:01 CD4^+^ T-clone (Figure 3c,d). No significant difference in sensitivity to lysis was observed between ALL Dg and relapses. Note that in addition to the molecular HLA typing previously described, we checked with MFC that ALL cells maintained HLA-A2 and HLA-DPB1*04:01 expression (Figure 2b,c). 

## 4. Discussion

For the first time, we report here an analysis of the T-cell-mediated lysis sensitivity of two successive ALL relapses that occurred after three immunotherapies (rituximab, geno-identical allogeneic HSCT, and anti-CD19 CAR T-cells). We showed that ALL relapses remained sensitive to all cytotoxic T-cell effectors tested, which recognized ALL cells through different receptors (CAR, TCR, and CD16). Even if CD19 targeted antigen expression decreased in relapse samples, CAR-T cell-mediated lysis remained effective.

We did not identify the appearance of surface molecules that may impact CAR-T cell persistence. Meanwhile, some unexpected targetable antigens, not routinely tested, appeared between Dg and the successive relapses, including FLT3 (CD135) or BAFF-R (CD268). Overall, our results provide options for antibody- or cell-based immunotherapy in the setting of post-CAR-T CD19+ relapses. 

The loss of CAR-T cells persistence is the first factor associated with CD19^+^ ALL re-occurrence after anti-CD19 CAR-T cells. It is now well-known that the co-stimulatory domain impacts CAR-T persistence, as the CAR construct with 4-1BB is associated with longer persistence than the CD28 domain [4]. The patient described here received tisagenlecleucel (a 4-1BB anti-CD19 CAR construct). Because relapses remained sensitive to CAR-T cell-induced lysis, a second injection of tisagenlecleucel could therefore have been discussed. Nevertheless, published data from second anti-CD19 CAR-T cells infusions report disappointing results, mostly due to lack of expansion [6]. An unfavourable in vivo immune environment may also impact CAR-T expansion and persistence. The murine scFv of CAR could lead to immune rejection, and some groups are experimenting use of humanized or fully human CAR constructs [14,15]. In addition, lack of co-stimulatory molecules (CD80, CD86, ICAM-1) and expression of molecules leading to immune cell anergy (CD47, HLA-II or PD-L1) through leukaemic cells or the bone marrow microenvironment have also been described [16,17,18,19,20]. Therefore, pilot studies combining a second injection of CAR-T and a PD1 blockade were conducted, with encouraging results [7,8,9], and a Phase I/II study is ongoing (CAPTiRALL EUDRACTN: 2021-003035-28). Other strategies targeting pathways known to contribute to T-cell exhaustion were also explored with encouraging results (CTLA-4, TIM-3, TGF receptor) [21]. In our case, we did not find a lack of co-stimulatory molecules or expression of checkpoint inhibitors. Nevertheless, we identified the appearance of CD14^+^/HLA-DR^lo/neg^ monocytic cells in the R3 sample. Retrospectively, we noted that this monocytic population had appeared in peripheral blood since day 4 after tisagenlecleucel infusion. These monocytic CD14^+^/HLA-DR^lo/neg^ cells recently emerged as tumour-induced immunosuppression mediators and are associated with poorer CAR-T expansion during manufacturing [22,23,24]. Targeting this immunosuppressive monocyte population might therefore have improved responses to a potential second anti-CD19 CAR-T injection in this patient.

Our analysis of broad immunophenotyping revealed newly appeared targetable antigens in relapse samples and led to unexpected therapeutic options. For this patient, anti-FLT3 targeted therapy could thus have been discussed in a curative or pre-emptive setting. In addition, we also showed that ALL cells remain sensitive to ADCC-mediated lysis (antibody + mCD16-T lymphocytes). The mCD16-T lymphocytes are described as universal CAR-T cells because their construction allows targeting of multiple antigens when used in combination with different antibodies. These immunotherapeutic T-cells could be helpful for targeting both one or more leukaemic antigens and the immunosuppressive microenvironment. 

In conclusion, our analysis of this refractory ALL revealed that, even in relapses, blastic cells remain sensitive to three-way T-cell-induced lysis. Moreover, we identified the appearance of new potential therapeutic targets in relapse samples. In light of our results, a second injection of anti-CD19 CAR-T could have been discussed. However, targeting the immunosuppressive microenvironment should probably have been considered. CD16-T-cells, seem particularly interesting in the post-conventional anti-CD19 CAR-T-cells relapse setting, allowing the simultaneous targeting of several antigens, possibly identified by broad immunophenotyping.

## Figures and Tables

**Figure 1 biomedicines-11-00345-f001:**
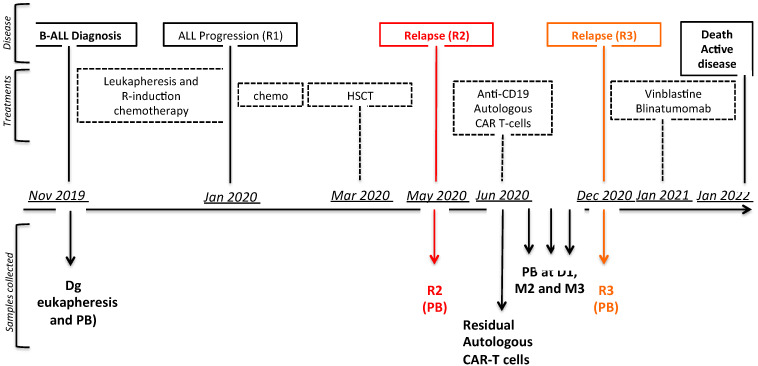
Clinical case medical history: disease progression, treatments received, and samples collected (Dg; R2; autologous CAR-T cells, blood samples and R3); ALL: acute lymphoblastic leukaemia; R: rituximab; HSCT: haematopoietic stem cell transplantation; PB: peripheral blood; chemo: chemotherapy.

**Figure 2 biomedicines-11-00345-f002:**
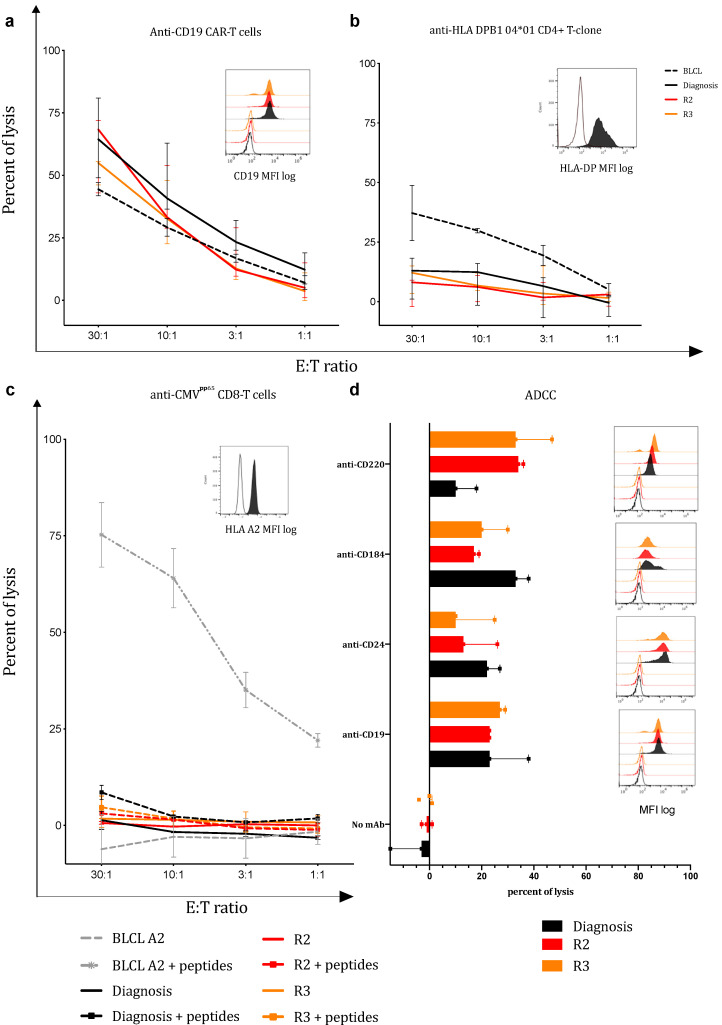
4h-^51^Cr cytotoxic assays over 4 h. Percentage of ALL lysis at four E:T ratio after co-culture with the autologous anti-CD19 CAR-T cell preparation (**a**); or anti-HLA DPB1*04:01 CD4+ T-clone (**b**); or anti-CMVpp65 CD8 T-cells (**c**); or through the ADCC pathway using anti-CD220, anti-CD184, anti-CD24, and anti-CD19 antibodies in combination with mCD16-T lymphocytes (**d**). Expressions of targeted antigens are, respectively, represented as median of immunofluorescence (MFI) in a log of 10. (BLCL: B cell lineage used as controls). Three experiments in triplicate were performed. Results are given as median/error.

**Figure 3 biomedicines-11-00345-f003:**
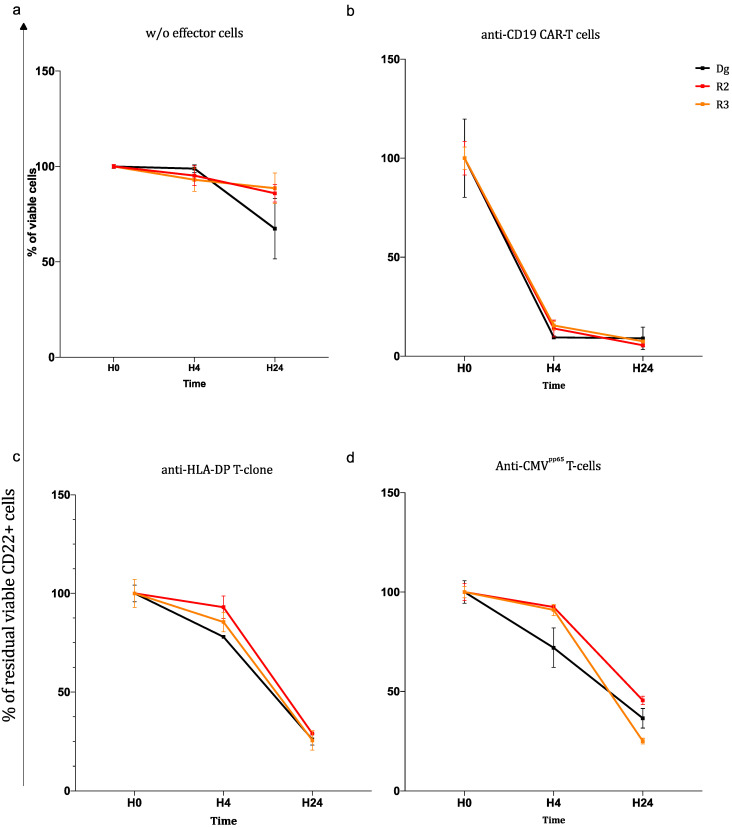
Long-term killing assays. Residual viable ALL cells at 3 time points of co-culture H0 (starting point), after 4 h (H4) of co-culture and after 24 h (H24) of co-culture: (**a**) with no effector, results showed total viable cells; (**b**) with a purified autologous anti-CD19 CAR-T cell preparation (percent of residual CD22+ viables cells); (**c**) with anti-HLA DPB1*04:01 CD4+ T-clone (percent of residual CD22+ viables cells); (**d**) with anti-CMVpp65 CD8 T-cells (percent of residual CD22+ viables cells). Two experiments in duplicate were performed at effector/ratio = 3:1. Results are given as mean + SD. CMVpp65 peptides: PepTivator CMVpp65 human, 130-093-435, Miltenyi Biotec; T eff: T-cell effectors, here polyclonal CD8+ T cells against CMV peptides.

## Data Availability

Data are available on request from the corresponding author.

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
