# Peer review of "Two Ways of Targeting a CD19 Positive Relapse of Acute Lymphoblastic Leukaemia after Anti-CD19 CAR-T Cells"

_biomedicines, 2023, doi:10.3390/biomedicines11020345_

Round 1
Reviewer 1 Report
Grain and colleagues evaluated the sensitivity of ALL cells for killing by various mechanisms (antibodies (ADCC, TCR, CART) at three different time points (diagnosis, after Rituximab and ASCT, and after CART). They show that ALL cells remain sensitive for these effectors at all three time points. In addition, by extensive immunophenotyping they show that new potentially targetable antigens become expressed during disease progression. The data are of interest and the study seems well performed.
My suggestions/comments are:
- How was MRD evaluated in this patient? What method and what sensitivity? It would be informative if these details are added to the paper.
- The authors uses cells after Ficoll density centrifugation. It is however not clear whether cells were used immediately thereafter (fresh) or after freezing and thawing, this information should be added. Was the percentage ALL cells in the used cell suspensions similar or in the same order of magnitude?
- Of the CD3+ T-cells, 11% were CART according to line 96 but 9% according to line 101? Please clarify.
- CART cells were 72% CD4 and 17% CD8+, what was the remaining 11%?
- The authors performed extensive immunophenotyping using LegendScreen. It would be informative if the details of this approach and panel are provided in a Supplement. Authors indicate that cells were first incubated with CD19 to obtain the leukemic blasts. Do the authors mean that they sorted the ALL cells or did they only use CD19 for gating?
- It would be informative of some of the immunophenotyping data are shown in some more detail, for example in the supplement. Especially more information of "classical" B-cell maturation markers like CD34, CD10, CD38, TdT, CD45 etc may be informative.
- The authors use Relative Fluorescence intensity. It will be helpful if the authors indicate (e.g. in the supplement) how this is calculated.
- The monocytic cells (CD14+/HLADR-) were detected at R3 and 7 days post CART. Is this population also detected in normal blood and was it detected at earlier time points (like diagnosis and R2)? If not detected at those time points, is that because these cells were not yet there or because there were many ALL cells that overgrow the monocytic cells?
- Figure 2: How are the data presented? Mean and SD? In how many replicates was the experiment performed? In Figure c the various black lines are very hard to distinguish from each other and there seem to be three dotted lines in the figure but only two in the legend. Please clarify. What is the black dotted line showing lysis? In the legend also the dotted orange and red lines should be added. In figure D please also indicate what data are shown (mean, median, range, SD, CV?) and indicate number of replicates. The histograms in figure d are too small for easy reading.
- Figure 3, 24 h co-cultures: I find it remarkable that without effector cells 100% of cells remain alive. I there no spontaneous cell death in such culture?
Reviewer 2 Report
Audrey Grain and colleagues present a quality and well-written brief report manuscript focused on two ways of targeting a CD19 positive relapse of acute lymphoblastic leukaemia after anti-CD19 CAR-T cells.
Authors analysed broad immunophenotyping and cytotoxicity assays, conducted on ALL cells from successive relapses in one patient after three different immunotherapies, in order to consider the optimal therapeutic way in this particularly complex setting.
Authors explored the immunophenotyping and lysis sensitivity of CD19+ ALL relapse after anti-CD19 CAR-T cells, and proposed different therapeutic options for such a high-risk disease. For that authors collected cells from successive B-ALL relapses from one patient. A broad immunophenotype analysis was performed. 51Cr cytotoxic assays, and long-term killing assays were conducted using T-cell effectors that are capable of cytotoxicity through 3 recognition pathways: antibody-dependent cell-mediated cytotoxicity, anti-CD19 CAR-T, and TCR.
Authors found that previously targeted antigen expression, even if maintained, decreased in relapses, and new targetable antigens appeared. Cytotoxic assays showed that ALL relapses remained sensitive to lysis mediated either by ADCC, CAR-T or TCR, even if the lysis kinetics were different depending on the effector used. They also identified an immunosuppressive monocytic population in the last relapse sample that may have led to low persistence of CAR-T.
They also found that CD19+ relapses of ALL remain sensitive to cell lysis mediated by T-cell effectors. In case of ALL relapses after immunotherapy, a large immunophenotype will make possible new therapies for controlling such high risk ALL.
Finally, authors conclude that their analysis of this refractory ALL revealed that, even in relapses, blastic cells remain sensitive to three-way T-cell-induced lysis. Moreover, they identified the appearance of new potential therapeutic targets in relapse samples. In light of their results, a second injection of anti-CD19 CAR-T could have been discussed. However, targeting the immunosuppressive microenvironment should probably have been considered. CD16-T-cells, seem particularly interesting in the post-conventional anti-CD19 CAR-T-cells relapse setting, allowing the simultaneous targeting of several antigens, possibly identified by broad immunophenotyping.
Overall, the manuscript is valuable for the scientific community and should be accepted for publication after edits are made.
===========================
Other comments:
1) Please check for typos throughout the manuscript.
2) Authors are kindly encouraged to cite the following review that describes various critical aspects defining functional activity of CAR-T cells.
DOI: 10.3390/cancers14041078
Author Response
Reviewer 2 :
- Please check for typos throughout the manuscript.
We agree. Typos of legends of the figure were corrected and standardized.
- Authors are kindly encouraged to cite the following review that describes various critical aspects defining functional activity of CAR-T cells.
DOI: 10.3390/cancers14041078
We have taken this remark into account. This review was cited and improved our discussion.
